



# Assessing ecohydrological separation in a northern mixed forest biome using stable isotopes

Jenna R Snelgrove[1], James M Buttle[2], Matthew J Kohn[3], Dörthe Tetzlaff[4,5]

[1]Environmental and Life Sciences Graduate Program, Trent University, Peterborough, ON, K9L 0G2, Canada
5   [2]School of the Environment, Trent University, Peterborough, ON, K9L 0G2, Canada
[3]Department of Geoscience, Boise State University, ID, 83725-1535, USA
[4]Leibniz Institute of Freshwater Ecology and Inland Fisheries (IGB), Berlin, Germany
[5] Department of Geography, Humboldt-University Berlin, Germany

*Correspondence to*: James M Buttle (jbuttle@trentu.ca)

10   **Abstract.** In recent years, much attention has been paid to the issue of ecohydrological separation during water uptake by vegetation. This has been spurred in part by the "two water worlds" hypothesis, whereby mobile "blue water" contributes to groundwater recharge and streamflow generation whereas less mobile "green water" held in the soil is taken up and transpired by vegetation. This study examines the potential for ecohydrological separation in a northern mixed forest in Ontario, Canada. Stable isotopic compositions of gross precipitation, bulk soil water and xylem water were measured throughout the 2016 growing season for four species: eastern white cedar, eastern hemlock, red oak and eastern white pine. Near-bole soil water contents and mobile soil water isotopic compositions were measured for the last three species. Mobile soil water did not deviate significantly from the local meteoric water line (LMWL); in contrast, both bulk soil water and xylem water deviated significantly from the LMWL, with xylem water significantly depleted in $^{18}$O and particularly $^2$H relative to bulk soil water. Near-surface bulk soil water experienced evaporative enrichment from pre-leaf out to peak leaf out under all tree canopies. There were inter-species differences in displacement of xylem water isotopic compositions from the LMWL and their temporal changes during the growing season, with those of coniferous species becoming isotopically enriched while those of red oak became more depleted in $^2$H and $^{18}$O. These divergences occurred despite thin soil cover (generally < 0.5 m depth to bedrock) which would constrain inter-species differences in tree rooting depths in this landscape. Our results failed to support the ecohydrological separation hypothesis that a distinct soil water source contributes to plant water uptake. Potential explanations for inter-specific differences in xylem water isotopic composition and its temporal evolution during the growing season in this northern forest landscape are assessed.

## 1 Introduction

Northern forest landscapes are highly sensitive to climate change (Laudon et al. 2017, Sprenger et al. 2018a) and may experience marked hydrological shifts in the future, such as changes in the amount, form and timing of precipitation (Carey et al. 2010, Hartmann et al. 2013) as well as increases in drought frequency and intensity (Brinkmann et al. 2019). Alterations in





snow accumulation and ablation have important implications for soil water availability to plants at the start of the growing season (Smith et al. 2011, Carey et al. 2013), and vegetation in northern landscapes can exhibit rapid responses to such changes (e.g. Myers-Smith et al. 2019). Understanding how northern forests may respond to these anticipated hydrological changes would benefit from greater knowledge of the sources of water taken up by major tree species in these landscapes (Guswas et

al. 2020). Environmental isotopes have often been used to study water use by vegetation (e.g. Evaristo et al. 2015), and efforts to account for the isotopic composition of plant water in relation to that of major water pools in forest landscapes has led to the ecohydrological separation or "two water worlds" hypothesis (McDonnell 2014). This hypothesis proposes that a highly mobile pool of soil water similar in isotopic composition to precipitation contributes to groundwater and streamflow while a less mobile pool of evaporatively-enriched soil water supplies plant transpiration (Goldsmith et al. 2012, Sullivan et al. 2016,

Knighton et al. 2019, Sprenger and Allen 2020).

McCutcheon et al.'s (2017) review of the ecohydrological separation hypothesis presented three assumptions that must be met for the hypothesis to be supported:

1. There is a distinct difference between the isotopic composition of water taken up by plant roots and the water that drains through the soil profile.

2. This difference can be linked to isotopically distinct soil water sources.

3. These isotopically distinct soil water sources arise from differences in soil water mobility.

Studies have called one or more of these assumptions into question, such as the assumption that mobile and tightly retained subsurface waters are independent water pools (Sprenger et al. 2018b), and there is mounting evidence that the two water worlds hypothesis is overly simplistic (Penna et al. 2018). For example, Bowling et al. (2017) noted that the assumption that

plants take up more strongly held soil water in the presence of less strongly retained soil water near the plant roots violates currently physiological understanding of how plants take up water, which is primarily driven by a potential gradient between the soil and the plant leaf or needle.

Nevertheless, the interface between soils and plants represents the potential source of novel advances in process understanding in ecohydrology, and systematic assessments of plant – soil water isotopic dynamics need to be examined across

distinct soil types and vegetation structures (Dubbert and Werner 2019). There is a particular need to examine relationships between the isotopic composition of xylem water in relation to that of potential source waters in northern forests (Tetzlaff et al. 2015, Penna et al. 2018), since much previous research into water use by vegetation using environmental isotopes has focused on tropical, seasonally dry or arid regions (Evaristo et al. 2015, Gaines et al. 2016). Most ecohydrological separation studies have also been restricted to the growing season (Liu et al. 2020), and greater consideration should be paid to the

seasonal variability of soil and plant water isotopic composition (McCutcheon et al. 2017, Sprenger et al. 2018a). This variability in northern landscapes is driven in part by a pronounced annual cycle that ranges from isotopically-depleted snowfall to isotopically-enriched summer rainfall (Birks and Gibson 2009), with important implications for the isotopic composition of source water available for plant uptake at the start of the growing season (McCutcheon et al. 2017, Allen et al. 2019). Plant – soil water isotopic dynamics may also differ between tree species in northern landscapes. Trees cannot be treated as "simple





transport vessels, or straws" (Evaristo et al. 2019, p 18), and inter-specific differences in the interplay between rooting depth and architecture and water flowpaths and storage in the soil profile may manifest themselves in the resulting isotopic composition of plant water uptake (Geris et al. 2015, Allen et al. 2019).

The purpose of this study is to examine the co-evolution of the isotopic composition of xylem water and soil water throughout the growing season for some common tree species in Canada's northern forest landscapes. We address the
following questions:

1.  What are the temporal changes in the isotopic composition of soil water and xylem water throughout the growing season in a northern forest landscape, and do the trajectories of such changes differ between tree species?

2.  Is there evidence for ecohydrological separation in a northern forest landscape, and does the potential for ecohydrological separation differ between tree species?

Answering these questions may improve our understanding of relationships between soil water and water taken up for transpiration by different tree species in northern forests and provide insight into how these species may respond to hydroclimatic change in northern landscapes.

## 2 Study Area and Methods

### 2.1 Study Area

The study was conducted in the Plastic-1 (PC-1, 23.3 ha) sub-catchment of Plastic Lake (Fig. 1) on the southern edge of the Canadian Shield near Dorset, Ontario, Canada (45°11' N, 78°50' W). Pleistocene glacial till overlies Precambrian metamorphic silicate bedrock (Wels et al. 1990), and thin soil cover is formed from sandy basal tills with an average depth of ~0.4 m to bedrock (Neary et al. 1987, Watmough et al. 2007). Visual observations of outcrops in PC-1 suggest the bedrock is relatively unfractured. Soils are overlain with a ~5 cm thick LFH layer (Neary et al. 1987) and are sandy with minor clay and
low organic matter contents showing little decline with depth (Buttle and House 1997). Forest cover is largely coniferous and dominated by red oak (*Quercus rubra*, Or), eastern white pine (*Pinus strobus*, Pw), eastern hemlock (*Tsuga canadensis*, He), white cedar (*Thuja occidentalis*, Ce), and black spruce (*Picea mariana*). The latter is confined to a wetland occupying the central portion of PC-1. Leaf-out of Or is in mid-May while senescence occurs by early October. A meteorological station ~500 m from the study site (Fig. 1) operated by the Dorset Environmental Science Centre (DESC) provides temperature and
precipitation data. Daily average temperatures range between -10° C and 18° C throughout the year, based on meteorological station data between 1981 and 2010. Mean annual precipitation is ~799 mm y$^{-1}$ of rain and ~260 mm y$^{-1}$ of snow water equivalent.





## 2.2 Gross rainfall sampling and potential evapotranspiration estimation

Gross precipitation ($P_g$) was measured weekly for amount and isotopic composition from May 27 to October 21, 2016 using a
bulk collector at the meteorological station which minimized isotopic fractionation via air exchange with the external
environment by reducing the water surface exposed to the atmosphere (Gröning et al. 2012). Snowmelt samples were obtained
from a snowmelt lysimeter located at Paint Lake, ~12 km northwest of PC-1 (Lane et al. 2020). Daily potential
evapotranspiration ($PET$) values were taken from Sprenger et al. (2018a), based on meteorological station data and the Penman-
Monteith equation (Allen et al. 1998).

## 2.3 Xylem water sampling

Four tree species in PC1 were selected to conduct xylem water sampling: Ce, He, Or and Pw. Five mature trees with similar
diameter at breast heights (DBH) were chosen for each species (Table 1). Sampled Or and Pw trees were intermixed, while He
trees were ~100 m away from the Or/Pw stand and Ce trees were ~200 m from the He trees and ~130 m from the Or/Pw stand
(Fig. 1). Xylem water was sampled six times between October 2015 to November 2016, including post-senescence 2015
(October 26 to November 3, 2015), pre-leaf-out 2016 (April 26 to April 29, 2016), post-leaf-out 2016 (June 20 to June 22,
2016), peak-leaf-out 2016 (August 8 to August 10, 2016), pre-senescence 2016 (September 23 to September 24, 2016), and
post-senescence 2016 (November 2 to November 4, 2016). Xylem cores were extracted from each tree at breast height using
an increment borer (3-thread, 5.15 mm core). Bark was removed from retrieved cores which were immediately stored in 200
mL glass scintillation vials with zero headspace. These were taped, sealed with Parafilm, and stored in a freezer to prevent
exchange with the atmosphere. Elapsed time between core extraction and storage in the sealed vials was on the order of 1
minute.

## 2.4 Soil water isotopic sampling

Concurrent bulk soil samples were obtained in a randomized direction 1 m from the bole of each tree sampled for xylem water.
Following litter layer removal, a minimum of 40 g of soil was collected using an auger at 5 cm depth increments until bedrock
was reached. Samples were double bagged in Ziploc bags while minimizing any stored air and stored at 4°C prior to analysis.
This bulk soil water was assumed to represent all water stored within the soil, including both mobile and more tightly held soil
water.

Mobile soil water was sampled from tension lysimeters installed at 0.1 and 0.4 m depths at 0.1 and 1 m from the tree bole
in a randomized direction for three He, three Or and three Pw trees that were not used for xylem water sampling (Snelgrove et
al. 2019). Lysimeters were sampled weekly between June 2and October 21, 2016 and re-set to a minimum negative air pressure
of 60 kPa using a hand pump. Samples were stored in sealed glass vials with zero headspace at 4°C prior to isotopic analysis.




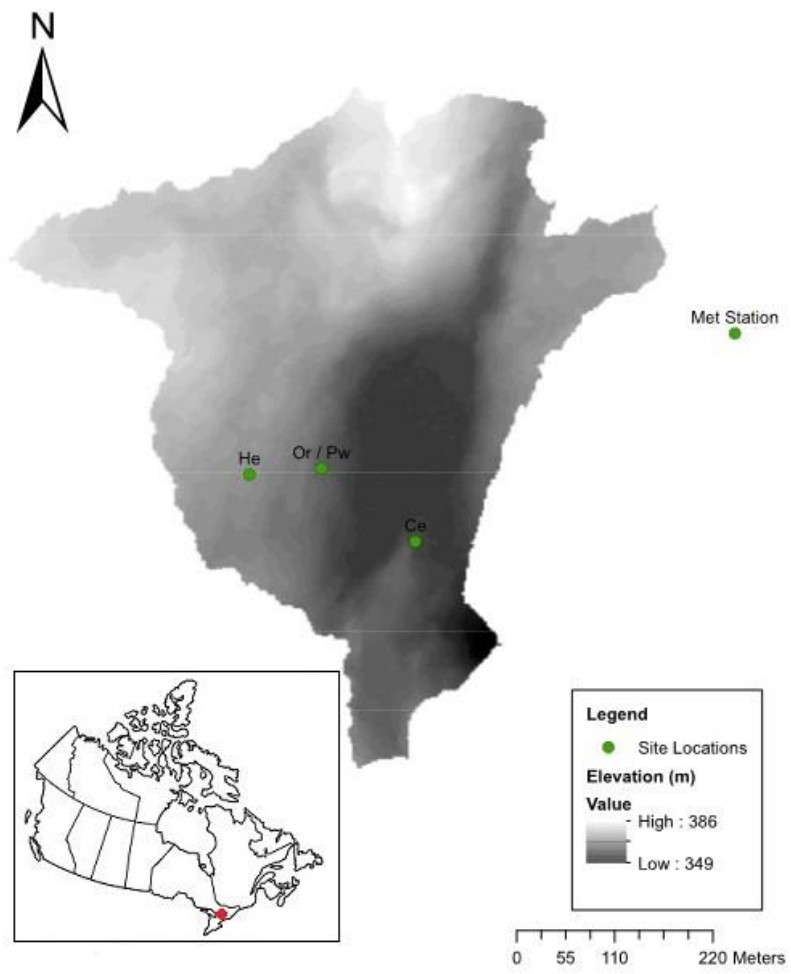

**Figure 1: A digital elevation model of the PC-1 catchment, showing the location of trees sampled for xylem water and bulk soil water**
**(Ce – eastern white cedar, He – eastern hemlock, Or – red oak, Pw – eastern white pine) and the meteorological station.**

Soil water content ($SWC$) was measured at two ATL-1 access tubes (http://www.delta-t. co.uk, last accessed May 30, 2019)
installed 0.1 and 1 m from the bole in a randomized direction for each of the three trees of a given species sampled for mobile
soil water. Measurements were concurrent with lysimeter sampling. A Delta T PR2/6 Soil Moisture Profile Probe$^{TM}$ measured
$SWC$ at each access tubes at 0.1-, 0.2-, 0.3-, and 0.4-m depths. Measurements at each depth were made three times per access
tube and averaged to obtain mean $SWC$ at each depth.





**Table 1. Tree height (m), diameter at breast height (DBH) (cm) and projected crown area (PCA) (m$^2$) for all eastern white cedar (Ce), eastern hemlock (He), red oak (Or) and eastern white pine (Pw) trees sampled for bulk soil water, xylem water, soil water content and mobile soil water. Soil surrounding trees indicated in italics was sampled for soil water content and mobile soil water.**

| Sampling Tree | Height (m) | DBH (cm) | PCA (m$^2$) |
|---|---|---|---|
| Ce-01 | 11.2 | 22.6 | 9.1 |
| Ce-02 | 10.9 | 25.5 | 9.1 |
| Ce-03 | 11.5 | 25.1 | 10.5 |
| Ce-04 | 8.5 | 21.3 | 7.8 |
| Ce-05 | 11.1 | 26.6 | 8.6 |
| *He-01* | *17.5* | *34.1* | *39.0* |
| *He-02* | *13.5* | *40.4* | *87.4* |
| *He-03* | *18.5* | *41.2* | *62.2* |
| He-04 | 17.8 | 35.2 | 63.6 |
| He-05 | 16.9 | 39.5 | 70.9 |
| *Or-01* | *22.3* | *50.6* | *21.4* |
| *Or-02* | *20.5* | *59.5* | *44.8* |
| *Or-03* | *17.8* | *66.5* | *107.5* |
| Or-04 | 13.6 | 50.8 | 75.4 |
| Or-05 | 19.4 | 57.9 | 111.2 |
| *Pw-01* | *17.6* | *60.5* | *21.0* |
| *Pw-02* | *30.1* | *62.4* | *12.7* |
| *Pw-03* | *26.8* | *53.2* | *78.5* |
| Pw-04 | 31.2 | 51.2 | 52.2 |
| Pw-05 | 20.3 | 47.4 | 25.1 |

## 2.5 Isotopic analyses

All isotope ratios are expressed relative to Vienna Standard Mean Ocean Water- Standard Light Antarctic Precipitation (VSMOW-SLAP, Coplen et al. 2002) using standard ‰ notation. Tree core samples were analyzed at the Boise State University Stable Isotope Laboratory. Xylem water was obtained from the cores by cryogenic extraction, followed by mass spectrometry using a Thermo Delta V Isotope Ratio Mass Spectrometer (IRMS) coupled with Thermo TC/EA configured for water injection analyses (Koeniger et al. 2011), with a precision of ±1.0 ‰ for $\delta^2$H and ±0.1 ‰ for $\delta^{18}$O. An extreme maximum limit on external reproducibility can be estimated from the compositional consistency among analyses of the same species on a single sampling date (e.g., all Or data on 4/26/16, etc.). This limit is a maximum because different trees are expected to have different compositions. The mean and median reproducibilities for these data are ~10‰ in $\delta^2$H and ~1‰ in $\delta^{18}$O. Bulk soil





water, tension lysimeter and $P_g$ samples were analyzed at the University of Saskatchewan using Los Gatos Research Liquid Water Off-Axis Integrated-Cavity Output Spectroscopy (Off-Axis ICOS) with a precision of $\leq \pm 1.0$ ‰ for $\delta^2$H and $\pm 0.2$ ‰ for $\delta^{18}$O. Bulk soil samples were analyzed using vapour extraction of water in an equilibrium state from the sealed Ziploc bags. Snowmelt samples were analyzed at the University of Toronto using a Los Gatos Research DLT-100 liquid water isotope analyser with a precision of $\leq \pm 1.0$ ‰ for $\delta^2$H and $\pm 0.12$ ‰ for $\delta^{18}$O.

The local meteoric water line (LMWL) was determined by regressing $\delta^2$H on $\delta^{18}$O for all snowmelt and $P_g$ samples (Klaus et al. 2015). Isotopic compositions of soil water and xylem water samples were compared with the LMWL using the line-conditioned excess (lc-excess), which defines the degree of deviation from the LMWL using:

$$lc - excess = \delta\ ^2H - a \times \delta^{18}O - b \tag{1}$$

where *a* and *b* are the LMWL's slope and intercept, respectively (Landwehr and Coplen 2006). Negative lc-excess indicates evaporative enrichment relative to the LMWL (Landwehr and Coplen 2006). McCutcheon et al. (2017) noted the benefit of lc-excess values in showing "isotopic distinction" between two water samples. These may differ markedly in their δD and $\delta^{18}$O values but can be considered to be genetically similar if they both plot on the LMWL.

Best-fit regression relationships were determined for xylem water in dual isotope space and the intersection points of these lines with the LMWL were obtained. Intersection points have been used to estimate the isotopic composition of the source precipitation that then may be altered as it undergoes fractionation by processes such as evaporation (Evaristo et al. 2015).

### 2.6 Statistics

All statistical analyses were performed using the *stats* package in R Statistical Software (R Core Team 2019). Shapiro-Wilks tests were used to assess normality of xylem water lc-excess values for each sampling period and species. One-way ANOVAs were used to compare differences in xylem water lc-excess between sampling periods for each tree species. Tukey HSD tests were then performed to identify significant differences in the data for each tree species. Inter-specific differences in xylem water lc-excess for a given sampling period were assessed using t-tests (unequal variances, Bonferroni-corrected).

### 3 Results

### 3.1 Hydrometeorological conditions during the sampling period

Precipitation data for the PC-1 meteorological station were not available for the Fall 2015 period. Total precipitation at the station for January 1 to October 31, 2016 (957 mm) exceeded the 30-year normal precipitation for January to October (854 mm) at the nearby Dorset MOE climate station (station i.d. 6112072). This was the result of above-average precipitation (largely as snow) in February and March, and a wetter-than-normal August (173 mm/mo vs. 76 mm/mo). Conversely, September 2016 was much drier than normal (47 mm/mo vs. 114 mm/mo). Daily PET ranged from 1 to 6.8 mm/day, peaking





in mid-June and declining to late October. Total PET from May 27 to October 21, 2016 was 597 mm, while total $P_g$ for the same period was 440 mm. The Canadian Drought Monitor (www.agr.gc.ca/atlas/maps_cartes/canadianDroughtMonitor/ accessed June 18, 2020) indicated that the 2016 growing season was relatively dry, with conditions ranging from abnormally dry (April) through to moderate (July to October) and severe drought (June).

### 3.2 Soil water contents

Total water depth in the upper 0.5 m of soil at 0.1 and 1 m from He, Or and Pw tree boles showed similar trends from early June 2016 to late October 2016 (Fig. 2): gradual draining through June into early August, a marked increase following 123 mm of rain between August 9 and August 17, and relatively high $SWC$s until the end of monitoring. $SWC$s were similar at 0.1 m and 1 m from the boles of trees of a given species. Greatest variability in $SWC$ was seen around Or trees, while the least was around He trees.

### 3.3 Isotope results

### 3.3.1 Precipitation

Figure 3a shows dual isotope plots of snowmelt and rainfall separated into periods prior to bulk soil water and xylem water sampling. Snowmelt samples represented pre-leaf out values, which were depleted in $^2$H and $^{18}$O relative to rainfall for the other periods; however, there was considerable overlap in rainfall isotopic composition with no clear demarcation between

sampling periods. The local mean water line (LMWL) using all samples was:

$$\delta\ ^2H = 7.0395 * \delta^{18}O + 4.6032;\ R^2 = 0.97 \tag{2}$$

Most throughfall and stemflow samples for He, Or and Pw trees fell on the LMWL, indicating limited isotopic enrichment of $P_g$ as it passed through the forest canopy (Snelgrove et al. 2019).





**Figure 2. Rainfall depths and range of total soil water depths held in the upper 0.5 m of soil at 0.1 (left-hand panels) and 1 m (right-hand panels) from the bole of He (second row), Or (third row) and Pw (fourth row) trees during the 2016 growing season. Vertical dashed lines indicate the timing of post-leaf out, peak leaf out and pre-senescence sampling of xylem water and bulk soil water.**



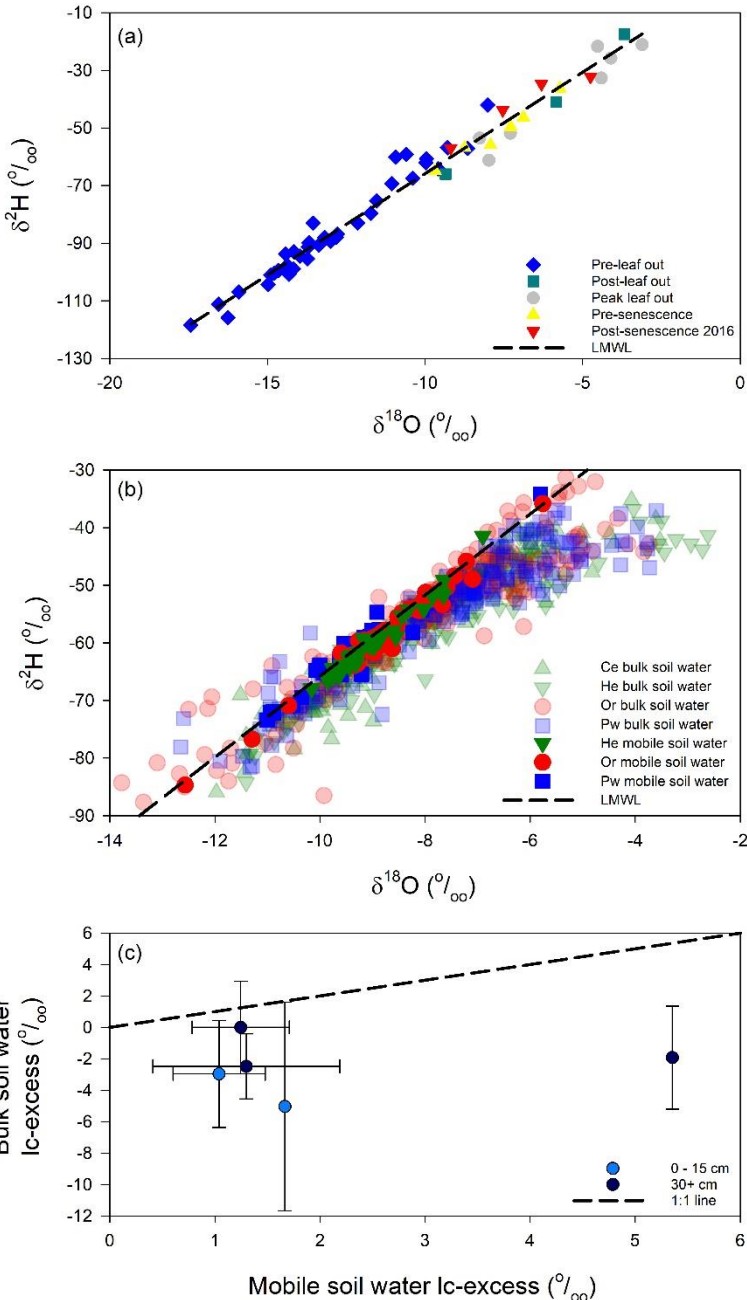

**Figure 3. Isotopic composition of snowmelt and rainfall for the various sampling periods for xylem water and bulk soil water and the estimated Local Mean Water Line (LMWL) (a); isotopic composition of bulk soil water and mobile soil water for the sampled tree species (b); mean (± 1 SD) bulk soil water lc-excess vs. mobile soil water lc-excess on or close to the same date of sampling for shallow (0 – 15 cm depth) and deep (≥ 30 cm depth) soil water (c).**



### 3.3.2 Mobile and bulk soil water

Mobile soil water fell on the LMWL (Fig. 3b), particularly deeper samples and those taken post-senescence in 2015 and 2016.

Snelgrove et al. (2019) also found limited evaporative enrichment of mobile soil water and weak correspondence between isotopic composition of mobile soil water and that of $P_g$, throughfall or stemflow inputs to the soil.

Most bulk soil water samples also plotted along the LMWL. However, the best-fit line of $\delta^2$H vs. $\delta^{18}$O for bulk soil water had a slope of 5.3, shallower than for meteoric water (7.0, Eq, 2), indicating evaporative enrichment of some samples. It is important to note the poor agreement between bulk soil water and mobile soil water lc-excess sampled within two days or less

of one another for a given tree species. Figure 3c plots bulk soil water samples from 0 – 15 cm depths against tension lysimeter samples at 10 cm depth, while bulk soil water samples from depths $\geq$ 30 cm are plotted against 40 cm depth tension lysimeter samples; bars indicate ±1 SD. Bulk soil water tended to be evaporatively enriched (more negative lc-excess) relative to mobile soil water sampled on or close to the same day.

Bulk soil water lc-excess values showed broadly similar distributions with depth beneath all tree species (Fig. 4). Post-

senescence 2015 samples showed considerable variability at a given depth and no obvious trends with depth, with a tendency for negative lc-excess values for all tree species. Lc-excess at all depths became more positive at pre-leaf out in late April, which may reflect large preceding inputs of snowmelt water which flushed the soil profile. Lc-excess was more negative at post-leaf out in late June, indicating evaporative enrichment. This was least pronounced for Pw relative to the other species. All species showed negative lc-excess for near-surface bulk soil water and increasing values with depth at peak leaf out in

early August. Lc-excess approached or equalled 0 at pre-senescence sampling in late September, consistent with decreased evaporation (Snelgrove et al. 2019). Lc-excess values at post-senescence 2016 were similar to those at post-senescence 2015 for Ce and to a lesser extent for Pw; however, 2016 values were generally more positive than 2015 values for He and Or.

### 3.3.3 Xylem water

Xylem water isotopic composition changed during the growing season, with the trajectory of this change differing between

species (Fig. 5). Coniferous species saw gradual enrichment of $^2$H and $^{18}$O from pre-leaf out to post-senescence in 2016. This transition was most pronounced for Ce and Pw, while He saw greater overlap in isotopic compositions of post-leaf out, peak leaf and pre-senescence samples. Xylem water for Or had a different temporal trajectory: both $^2$H and $^{18}$O became depleted from pre-leaf out to peak leaf out, followed by slight depletion of $^2$H from peak leaf out to pre-senescence, and slight depletion of $^{18}$O from pre-senescence to post-senescence 2016.




**Figure 4. Bulk soil water lc-excess at different depths for the sampling periods for Ce (first row), He (second row), Or (third row) and Pw (fourth row).**



There was little overlap of xylem water and bulk soil water in dual isotope space (Fig. 6), with the former having much more negative $\delta^2$H and to a lesser extent $\delta^{18}$O relative to bulk soil water. There were pronounced inter-species differences in

xylem-water lc-excess values and their relationship with near-surface soil water (Fig. 7). Bulk soil water lc-excess at 0-5 cm depth is shown, since near-surface soil experienced the greatest evaporative enrichment and thus the most negative lc-excess. Xylem water lc-excess values for a given sampling period and tree species were normally distributed. One-way ANOVA indicated no significant difference in xylem water lc-excess between sampling periods for He; conversely, other species showed significant differences between some sampling periods. There were no significant inter-species differences in xylem

water lc-excess post-senescence 2015; however, distinctions emerged during subsequent sampling periods (Fig. 7, Table 2). Lc-excess for Or xylem water was less negative compared to other species and showed considerable overlap between sampling periods, with the most negative values at pre-senescence. They also often overlapped near-surface soil water lc-excess. Lc-excess for He was similar for all sampling periods, although inter-tree variability declined progressively from pre-leaf out to pre-senescence. There was occasional overlap of xylem water and near-surface bulk soil water lc-excess values. A different

relationship occurred for Ce and to a lesser extent Pw. Xylem water lc-excess for the former became more negative from post-senescence 2015 to peak leaf out and then became more positive. Lc-excess for Ce was generally more negative than for other species and was often more negative than the most evaporatively-enriched bulk soil water. Lc-excess for Pw also declined from post-senescence 2015, becoming most negative at pre-senescence. Pw lc-excess also tended to fall outside the near-surface bulk soil water range, although there was more overlap than for Ce xylem water.

Figure 8 presents soil water – xylem water offsets for $\delta^2$H throughout the study period, defined as the difference between the mean isotopic composition of soil water surrounding a sampled tree and xylem water for that tree. Offsets for $\delta^{18}$O showed similar patterns to those for $\delta^2$H and are not shown. Temporal trajectories of these offsets showed inter-specific differences. Minimum offset values for Ce occurred at post-senescence in 2015. Values rose to maxima either at post-leaf out or peak leaf out before declining to post-senescence 2016. Offsets were more temporally-constant for He with maxima at

peak leaf out. There was a marked decline in the Or $\delta^2$H offsets from 2015 post-senescence to pre-leaf out, followed by a gradual increase to maxima at either pre-senescence or 2016 post-senescence. Pw had more temporally constant $\delta^2$H offsets with minima at either pre-leaf out or post-leaf out.

Best-fit equations for xylem water in dual isotope space and the intersection points of these lines with the LMWL are given in Table 3. The Or intersection point fell within the range of observed rainfall and snowmelt $\delta^2$H and $\delta^{18}$O (Fig. 3), suggesting

precipitation may have been the source of Or xylem water. This occurred to a lesser extent for Pw, whose intersection point fell at the lower end of rainfall and snowmelt isotopic compositions. However, Ce and He intersection points indicated no obvious precipitation source of xylem water from these trees.





**Figure 5. Xylem water isotopic composition for the different sampling periods for Ce (a), He (b), Or (c) and Pw (d). LMWL – Local Mean Water Line.**



**Figure 6. Isotopic composition of xylem water and bulk soil water for the sampled tree species. Ce – eastern white cedar, He – eastern hemlock, Or – red oak, Pw – eastern white pine, LMWL – Local Mean Water Line.**






**Figure 7. Box-and-whisker plots of xylem water (left-hand-side panels) and near-surface (0 – 5 cm depth) bulk soil water (right-hand-side panels) lc-excess values for Ce (first row), He (second row), Or (third row) and Pw (fourth row) for the different sampling periods. Horizontal line – median, box – 1st to 3rd quartiles, whiskers – maximum and minimum values. Xylem water box and whiskers with different letters for a given tree species are statistically different (p = 0.05) based on Tukey HSD tests.**






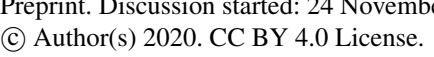

Figure 8. Bulk soil water – xylem water δ²H offsets (Δδ²H) for the sampled trees for each sampling period. See text for details on derivation of offsets. Positive values of Δδ²H indicate that xylem water δ²H is more negative than the corresponding mean bulk soil water δ²H.



**Table 2. Significant differences in mean xylem water lc-excess for a given sampling period, determined from t-tests with p = 0.0015 (equivalent to p = 0.05 following Bonferroni correction).**

| Post-senescence 2015 | Pre-leaf out | Post-leaf out | Peak leaf out | Pre-senescence | Post-senescence 2016 |
|---|---|---|---|---|---|
| - | Ce < Or | Ce < He | Ce < He | Pw < He | Ce < Or |
| | Ce < Pw | Ce < Or | Ce < Or | | Pw < Or |
| | Pw < Or | Pw < Or | Ce < Pw | | |
| | | | He < Or | | |
| | | | Pw < Or | | |

**Table 3. Xylem water best-fit lines in dual isotope space, and intersection points between best fit lines and the Local Mean Water Line (LMWL)**

| Tree species | Best-fit line | $R^2$ | p | Intersection point between xylem water best-fit line and LMWL | |
|---|---|---|---|---|---|
| | | | | $\delta^{18}O$ (‰) | $\delta^2H$ (‰) |
| Ce | $\delta^2H = -29.2 + 6.6*\delta^{18}O$ | 0.43 | < 0.001 | -56.0 | -399.6 |
| He | $\delta^2H = -21.9 + 6.0*\delta^{18}O$ | 0.66 | < 0.001 | -21.9 | -154.3 |
| Or | $\delta^2H = -29.7 + 4.9*\delta^{18}O$ | 0.53 | < 0.001 | -14.7 | -102.0 |
| Pw | $\delta^2H = -47.4 + 4.2*\delta^{18}O$ | 0.37 | < 0.001 | -17.2 | -120.0 |

## 4 Discussion

### 4.1 What are the temporal changes in the isotopic composition of soil water and xylem water throughout the growing season in a northern forest landscape, and do the trajectories of such changes differ between tree species?

Bulk soil water isotopic composition exhibited similar trends beneath the different tree species' canopies. Near-surface bulk soil water showed evaporative enrichment at peak leaf out, when *SWC*s reached a minimum. Total depth of soil water in the upper 0.5 m of soil at peak leaf out was ~50 mm (equivalent to a mean *SWC* of 0.1 $m^3 m^{-3}$), and lc-excess approached -30 ‰ beneath He (Fig. 4). Increased enrichment of bulk soil water relative to both the LMWL and corresponding mobile water with declining *SWC*s was also observed by Zhao et al. (2013) in a loamy soil and by Sprenger et al. (2018b) at PC-1. The bulk soil

water slope in dual isotope space at PC-1 (5.3) agreed with values of 5.3 to 5.8 reported by Bowling et al. (2017) for riparian soils in eastern Utah, while lc-excess values for near-surface soils were within the range given in McCutcheon et al. (2017) for a semi-arid landscape in southwest Idaho. Bulk soil water isotopic composition following senescence did not return to that of





the previous year for all tree species (Fig. 4). Brooks et al. (2010) also noted an inter-annual difference in soil water isotopic composition at the same location and depth for the same time of the year.

300 In contrast to similar trends in bulk soil water isotopic composition, xylem water showed inter-specific differences in its displacement from both bulk soil water and the LMWL, and the temporal trajectory of that composition throughout the growing season. The range in xylem water slopes in dual isotope space (4.2 to 6.6, Table 3) overlapped that for a mixed hardwood forest in Pennsylvania (6.5; Gaines et al. 2016) and were similar to those for hemlock (5.5 – 6.6) in central New York state (Knighton et al. 2019). Xylem water best-fit lines were displaced below both the LMWL and bulk soil water, with the least

305 displacement for Or and the greatest for Ce. Nevertheless, xylem water slopes overlapped that for bulk soil water (5.3), as observed by McCutcheon et al. (2017) but in contrast to Bowling et al.'s (2017) measurement of xylem water slopes (3.7 to 4.7) that were distinctly lower than the soil water $\delta^2$H vs. $\delta^{18}$O slope.

 While all sampled tree species had similar mean lc-excess post-senescence 2015, some species diverged from these initial values more than others. The greatest shifts in xylem water lc-excess were for Ce, with no significant change in mean lc-excess

310 between sampling periods for He. The more negative lc-excess for Ce at peak leaf out may explain why post-senescence 2016 lc-excess for Ce did not return to the post-senescence 2015 values, unlike for the other species (Fig. 7). White and Smith (2015) also found that xylem water $\delta^2$H at the end of the growing season approached but did not equal values at the initiation of dormancy in the previous year, suggesting xylem water does not necessarily reset to a consistent isotopic composition at the start of each growing season.

315 Importantly, inter-specific variations in xylem water isotopic composition and its temporal changes are likely not due to distinct environmental conditions for the different tree species, given the close proximity of the sampled trees and similar soil conditions under the tree canopies. They also contrast with some previous work. White and Smith (2015) saw insignificant seasonal variations for box elder (*Acer negundo* L.) or river birch (*Betula nigra* L.) as well as limited inter-specific differences at a given phenological stage in the foothills of the southern Appalachian Mountains. McCutcheon et al. (2017) found stem

320 water lc-excess from shrub (*Prushia tridentata*, *Prunus virginiana*, *Artemisea tridentata*), deciduous (*Betula occidentalis*, *Salix lucida*) and coniferous (*Pseudotsuga menziesii*, *Pinus ponderosa*) tree species either became less negative or remained stable during the growing season. Xylem water $^2$H for deciduous (beech and oak) species in Switzerland was more depleted compared to that of spruce (Allen et al. 2019), which was attributed in part to differences in elevation. This contrasts with results from PC-1, where conifers (particularly Ce and Pw) tended to have more depleted $^2$H than Or. Nevertheless, White and Smith (2015)

325 noted a divergence in xylem water $\delta^2$H between species from the beginning of the dormant period. This was similar to PC-1, where Or xylem water generally became more depleted moving from pre-leaf out to post-senescence 2016 while the reverse trend occurred for Ce, Pw and to a lesser extent He (Fig. 5). White and Smith (2015) suggested such divergence may be due to differing periods of inactivity for the two species they studied. This may be relevant to deciduous species such as Or that experiences leaf-out and senescence and whose timing and intensity of water use may differ from that of coniferous species.

330 Philips and Ehleringer (1995) found evaporatively enriched stem water in big-toothed maple (*Acer grandidentatum* Nutt.) and Gambel's oak (*Quercus gambelii*) before leaf flush and saw stem water move closer to the LMWL at full leaf out. We found





similar results for Or where lc-excess became less negative at peak leaf out relative to pre-leaf out, although the difference was
not statistically significant (Fig. 7). Gaines et al. (2016) noted xylem water from larger oak (*Quercus*) and hickory (*Carya*)
trees tended to be more depleted in heavy isotopes than that of smaller *Acer*. The reverse was the case at PC-1, where the
smallest trees (Ce) showed the most depleted xylem water $^2$H.

    Inter-specific differences and temporal changes in xylem water isotopic composition at PC-1 clearly exceeded inter-tree
differences for a given species. Standard errors for xylem water for a given species and sampling date ranged from 0.39 to
5.13 ‰ for $\delta^2$H and 0.06 to 0.79 ‰ for $\delta^{18}$O and were similar to previously reported results. Retzlaff et al. (2001) found
insignificant differences in xylem water $\delta^2$H between trees of a given species on a given measurement date, with standard
errors of 5 ‰ or less, while White and Smith (2015) found maximum standard errors for xylem water $\delta^2$H and $\delta^{18}$O of 5.49 ‰
and 0.84 ‰ for *Acer negundo* L. and 3.83 ‰ and 0.65 ‰ for *Betula nigra* L.

## 4.2 Is there evidence for ecohydrological separation in a northern forest landscape, and does the potential for ecohydrological separation differ between tree species?

We did not find evidence of ecohydrological separation for all tree species examined here. Ecohydrological separation depends
on some degree of correspondence between the isotopic composition of xylem water and that of a water store that can supply
transpiration. Such correspondence permits derivation of the soil depth from which trees abstract water, using approaches that
range from visual assessment of the depth of overlap of xylem water and soil water isotope values (e.g. Goldsmith et al. 2012)
to Bayesian linear mixing models (e.g. Evaristo et al. 2019). Such approaches could not be applied at PC-1 since xylem water
$\delta^2$H and $\delta^{18}$O often fell outside the range for bulk soil water. Pronounced soil water – plant water $\delta^2$H offsets observed for all
species at PC-1 (Fig. 8) may be a general occurrence in temperate forests (Barbeta et al. 2020). However, these values exceeded
the mean $\delta^2$H offset of 10.6 ± 3.05 ‰ reported by Barbeta et al. (2020) for potted saplings of European beech (*Fagus sylvatica*
L.). Barbeta et al. (2020) found these offsets to be most negative when soils were very dry, whereas the smallest offsets at PC-
1 were either at post-senescence 2015 (Ce) or pre-leaf out (He, Or, Pw) when *SWC*s would be relatively large.

    Nevertheless, there were inter-specific differences in the xylem water – bulk soil water relationship. Lc-excess of Or xylem
water frequently overlapped that of near-surface bulk soil water for the same or a preceding sampling period (Fig. 7),
suggesting ecohydrological separation. However, Or xylem water samples plotting well below bulk soil water in dual isotope
space (Fig. 6) indicates that support for the ecohydrological separation hypothesis was not as strong as that from previous work
(e.g. Goldsmith et al. 2012). There was no consistent overlap and often pronounced differences between xylem water and bulk
soil water isotopic compositions for the other species, indicating they did not demonstrate ecohydrological separation. The
greatest differences between xylem water and bulk soil water isotopic compositions for Ce and He were generally at peak leaf-
out (Fig. 8) following a protracted decline in *SWC*s (Fig. 2), and these differences persisted following soil rewetting. This
contradicts results from Evaristo et al.'s (2019) controlled experiment, where ecohydrological separation (indicated by





correspondence between the isotopic compositions of plant water and low mobility soil matrix water) was negligible under drought conditions and most marked at the transition from drought to rewetting.

Distinctions between xylem water and bulk soil water isotopic compositions have been noted elsewhere. Brooks et al. (2010) found some plant water plotting beyond (and generally below) the range of soil water $\delta^2H$ and $\delta^{18}O$ in dual isotope space but did not address it, with Goldsmith et al. (2012) later suggesting this xylem water may have undergone further evaporation. Of the 17 isotope-based studies of plant water use in temperate forests (similar to the PC-1 forest landscape) cited by Evaristo et al. (2015), 13 reported both soil water and xylem water offsets (analogous to lc-excess used here) and four

indicated xylem water offsets falling below the range (i.e. were more negative) of the corresponding soil water offsets. More recent work (e.g. Geris et al. 2015, White and Smith 2015, Bowling et al. 2017, Hervé-Fernández et al. 2016, McCutcheon et al. 2017, Brinkmann et al. 2019) also saw a distinction between the isotopic composition of xylem water and possible water sources that might support transpiration.

### 4.3 Why is xylem water isotopically distinct from bulk soil water?

In his argument for the "two water world" hypothesis McDonnell (2014) noted that "the isotopes don't lie" (p 2/17). Perhaps not, but neither do they supply us with an unambiguous explanation of their values in all circumstances. The question remains: why is xylem water isotopically distinct from bulk soil water, both at PC-1 and elsewhere? Several lines of argument have been put forward to account for this, some of which are more plausible than others in a northern mixed forest such as at PC-1.

1. Xylem water is accessing an un-sampled water source not captured in bulk soil water isotope values (White and Smith
2015). This assumption is favoured by the ecohydrological separation literature (McCutcheon et al. 2017). The possibility that bulk soil water does not represent root-absorbed water (McCutcheon et al. 2017) implies plants selectively access an isotopically-fractionated portion of soil water, which would be consistent with the ecohydrological separation hypothesis. However, soils in PC-1 at peak leaf out were very dry while Ce xylem water lc-excess was much more negative than corresponding bulk soil water (Fig. 7). Presence of sufficient soil water with very negative lc-excess
values that could both match the xylem water lc-excess and supply the tree's transpiration demand is unlikely under these circumstances.

    Rooting behaviour of trees studied at PC-1 is unknown, and the issue of root activity is one of the most difficult dilemmas facing plant ecology and ecohydrology (Beyer et al. 2016). Inter-specific differences in plant water lc-excess may reflect variations in rooting characteristics and shifts in the depth of water uptake in response to changes in water
availability (McCutcheon et al. 2017, Dubbert and Werner 2019). Nonetheless, shallow soils overlying bedrock at PC-1 would likely constrain the ability of different tree species to develop marked contrasts in rooting architectures (e.g. deep tap roots for some species vs. shallow root networks for others). Trees may have accessed water held in bedrock fractures that may be isotopically distinct from mobile soil water (Oshun et al. 2016). However, bedrock in PC-1 appears to be relatively unfractured, making it difficult to envisage sufficient water held in fractures that could supply transpiration to a





significant extent. This echoes Gaines et al. (2016), who saw little evidence that roots within or below fractured bedrock in central Pennsylvania were consistent major contributors to transpiration. Here, the root length densities of *Quercus, Carya, Pinus and Acer* species were highest in the soil's upper 10 cm and displayed a negative relationship with depth (Gaines et al. 2016). Assuming similar rooting behaviour in PC-1, this suggests transpiration would be largely supplied by near-surface soil water. While this water had the greatest degree of isotopic enrichment, xylem water lc-excess was often

more negative (Fig. 7). Thus, an unsampled water source is an unlikely cause of xylem water isotopic compositions in PC-1.

2.  Xylem water at a given time may be influenced by the isotopic composition of water taken up days or months beforehand (McCutcheon et al. 2017, Penna et al. 2018, Evaristo et al. 2019). Sprenger et al. (2018c) estimated median ages of total soil water storage in PC-1 ranging from 31 (25th percentile) to 74 days (75th percentile). This storage effect may assist in

explaining the frequent distinction between xylem water and bulk soil water lc-excess for a given sampling period, and is supported by partial (He, Pw) or complete (Or) overlap of the lc-excess of post-senescence 2015 soil water and pre-leaf out xylem water in 2016 (Fig. 7). However, Ce xylem water lc-excess was much more negative than post-senescence 2015 bulk soil water, suggesting this mechanism may differ in importance between tree species. Such a mechanism also fails to account for Ce and to some extent Pw lc-excess values that were much more negative than any bulk soil water

(Figs 4 and 7).

3.  There may be errors in the xylem water isotopic composition values. Miller et al. (2018) noted that cryogenic extraction (the approach used here to obtain xylem water samples) resulted in more depleted xylem water $^2$H in spring wheat compared to other extraction methods. However, such an experimental issue might be expected to influence measured xylem water isotopic compositions relatively consistently across species and seasons, and does not readily account for the

inter-specific differences in xylem water $\delta^2$H observed at PC-1. Others have found no consistent evidence that cryogenic extraction alters xylem water's isotopic values (Barbeta et al. 2015, Grossiord et al. 2017). Berry et al. (2017) suggested cavitation (air entry) during extraction of stem cores from trees and increasing time lags between extraction and sealing the sample allows evaporation that fractionates the remaining sample water. Given the rapid processing of xylem water samples at PC-1, the interspecific and temporal variation in xylem water isotopic composition and its pronounced

divergence from that of all other potential water sources, the displacement of xylem water from bulk soil water is likely not attributable to a methodological issue.

4.  Fractionation of soil water may have occurred during water uptake by trees. A key assumption of the ecohydrological separation hypothesis is that root water uptake does not alter the isotopic composition of water in the plant roots or stems (Goldsmith et al. 2012, Penna et al. 2018). Previous work has generally not observed fractionation during water uptake

by plants (e.g. Dawson et al. 2002, Vargas et al. 2017 and references therein), although Vargas et al. (2017) noted this may have resulted from saturated soil conditions used in many studies. There is increasing recognition that differences between xylem water and soil water isotopic compositions may result from isotopic fractionation induced by internal plant processes during water uptake (Berry et al. 2017). Ellsworth and Williams (2007) showed that 12 of 16 species of





woody plants demonstrated H isotope fractionation at the soil-root interface, while Vargas et al. (2017) found plants

discriminated against $^{18}$O and $^{2}$H during water uptake with differences between $\delta^{18}$O and $\delta^{2}$H in soil water relative to

plant water increasing with transpiration water loss. This would lead to more negative $\delta^{18}$O and $\delta^{2}$H in plant water

relative to soil water, as seen at PC-1. It also suggests the greatest differences between xylem water and soil water lc-

excess would be at peak leaf out, when PC-1 soils were at their driest (Fig. 2). This was the case for Ce which showed

clear separation between soil and xylem water lc-excess at peak leaf out (Fig. 6). This separation also occurred at pre-

senescence for Pw; however, there was overlap between soil water and xylem water lc-excess for Or and He. Thus, the

potential for fractionation during water uptake may be a major cause of deviations between soil water and xylem water

isotopic compositions and may differ between tree species in northern mixed forests.

5.  Water in the trees may have been fractionated following root uptake via evaporation, transpiration or other internal

mechanisms. Changes in the isotopic composition of xylem water relative to that of soil water has been attributed to such

processes as xylem-phloem exchange during water stress (Cernusak et al. 2005, Bertrand et al. 2014), isotopic depletion

of storage water in xylem tissue compared to water moving via conductive tissues (Barbeta et al. 2020), H fractionation

when water movement in the tree occurs predominantly via symplastic rather than via apoplastic pathways (Lin and

Sternberg 1993, Ellsworth and Williams 2007), and fractionation within the tree's leaves which then impacts the isotopic

composition of phloem sap (Farquhar et al. 2007). There is also the potential for evaporation through the tree's bark

(Dawson and Ehleringer 1991, Smith et al. 1997). Bowling et al. (2017) thought this could not explain evaporative

enrichment of xylem water in their study given the large stems sampled and removal of bark from all stem samples, as

was the case at PC-1. The potential for evaporative enrichment of xylem water can be assessed by considering the

isotopic composition of the precipitation from which the xylem water originated (Evaristo et al. 2015). Bowling et al.

(2017) argued that while the intersection point between an evaporation line and the LMWL is appropriate for estimating

the precipitation source water for surface or soil water undergoing evaporation, there is no analogous mechanism by

which plant xylem water can become similarly enriched. Nevertheless, the xylem water precipitation sources in Table 3

are instructive. Xylem water source $\delta^{2}$H and $\delta^{18}$O for Or and Pw were within the range of snowmelt and rainfall sampled

at PC-1 (Fig. 3) as well as the range of source precipitation $\delta^{2}$H reported for a variety of forest types (Evaristo et al.

2015). However, estimated source precipitation for He and particularly Ce was more isotopically depleted than any

snowmelt samples. Thus, it is difficult to attribute xylem water isotopic composition for some tree species to simple

evaporation of precipitation source water held within the tree.

6.  Fractionation may have resulted from interactions with carbonates or other geochemical and organic constituents

(Bowling et al. 2017), including isotopic effects between soil water and cations/clay minerals (Oerter et al. 2014), organic

matter (Orlowski et al. 2016), and rock-water interactions (Lin and Horita 2016, Oshun et al. 2016) that differ for H and

O. The potential for any or all of these processes to induce differences in xylem water isotopic composition relative to

soil water at PC-1 is not known, although the low clay contents of PC-1 soils make significant isotopic effects with clay



minerals unlikely (Sprenger et al. 2018b). Regardless, it would be reasonable to expect that such processes would be similar in the soil water surrounding the different tree species. Thus, they do not easily explain interspecific differences in the degree of isotopic displacement of xylem water from the LMWL and bulk soil water.


Despite considering these manifold possible processes, the cause of isotopic distinction between xylem water and potential water sources remains unclear. Thus, our inability to support the ecohydrological separation hypothesis for some tree species at PC-1 remains. Nevertheless, some mechanisms noted above (e.g. fractionation at the plant root) are more plausible under conditions at PC-1 than others (e.g. xylem water accessing an unsampled source of water not reflected in bulk soil water

isotope values, errors in determination of xylem water isotopic composition).

Regardless, our results provide novel insight into potential changes in hydroecological fluxes in northern mixed forests in response to hydroclimatic change. The landscape surrounding PC-1 in Ontario is projected to experience increases in summer temperatures above the 1971-2000 baseline of between 2 and 8°C (and presumably accompanying increases in evaporation) and decreases in summer rainfall of up to 25 mm for the 2071-2100 period (McDermid et al. 2015). The 2016 growing season

was a particularly dry one; thus, the small $SWC$s and associated inter-specific differences in xylem water – bulk soil water isotopic relationships presented here may become typical of future conditions in this and similar northern forest landscapes across the northern hemisphere.

## 5 Conclusion

We examined the co-evolution of xylem water and bulk soil water isotopic compositions during the growing season for four

tree species in a northern mixed forest. We also evaluated whether these species supported the ecohydrological separation hypothesis. The major findings are as follows:

1  Bulk soil water isotopic composition showed similar temporal changes below the canopies of all tree species, with evaporative enrichment of near-surface soil water from pre-leaf out to peak leaf out followed by a return to values along the LMWL at post-senescence.

485 2  In contrast, xylem water isotopic composition showed inter-specific differences in both the degree of its displacement from the LMWL and bulk soil water and the temporal trajectory of its changes from pre-leaf out to post-senescence.

3  The deciduous tree species (red oak) exhibited the greatest overlap of xylem water and bulk soil water isotopic compositions; nevertheless, several xylem samples that were isotopically distinct from bulk soil water indicated red oak showed only partial support for the ecohydrological separation hypothesis. Xylem water isotopic compositions for the

490    coniferous species (eastern white cedar, eastern hemlock and eastern white pine) showed marked and differing degrees of displacement from both the LMWL and bulk soil water, and thus did not support ecohydrological separation.

4  The temporal trajectory of xylem water from the deciduous tree species differed from that for the conifers. Red oak xylem water experienced depletion in both [2]H and [18]O during the growing season, while conifer xylem water showed





isotopic enrichment. This may be related to inter-specific variations in the timing and intensity of growing season water
495    use in northern mixed forests and requires further study.

5    A review of possible reasons for distinctions between xylem water and soil water isotopic compositions for these tree
species suggested that some mechanisms (e.g. fractionation at the tree root) were more plausible than others (e.g. an
unsampled source of water taken up during transpiration) when considered in the context of the study site's
characteristics. Nevertheless, there appeared to be inter-specific differences in the degree to which these mechanisms
may account for the varying relationships between xylem water and soil water isotopic compositions over the course of
the growing season in this mixed northern forest. These should be explored in further research as we move beyond the
ecohydrological separation hypothesis to attempt to understand more fully how trees take up water during transpiration in
order to predict their response to anticipated hydroclimatic changes in northern forest landscapes.

## 6 Data Availability

The underlying research data can be accessed at
https://dataverse.scholarsportal.info/dataset.xhtml?persistentId=doi:10.5683/SP2/TGCHV6

## 7 Acknowledgements

This work was funded through the Natural Sciences and Engineering Research Council of Canada (2015-06116) and the
European Research Council (ERC, Project GA 335910 VeWa). Thanks to the Dorset Environmental Research Centre for
meteorological data, Carl Mitchell (University of Toronto Scarborough) for snowmelt lysimeter isotope data, Samantha Evans
(Boise State University), Jeff McDonnell and Kim Janzen (University of Saskatchewan) for isotopic analyses, and Robert
Monico and Ciara Cooke for field assistance.

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
