# Peer review of "Co-evolution of xylem water and soil water stable isotopic composition in a northern mixed forest biome"

_Hydrology and Earth System Sciences, 2020_

## Referee Comment (RC1) · Anonymous Referee #1 · 23 Dec 2020

The study by Snelgrove et al. investigates if ecohydrologic separation was possible in a northern mixed forest in Ontario, Canada. Their study design is built to assess the co-evolution of mobile-, bulk soil- and xylem water isotopic compositions during the year 2016. They formulate two questions to be considered during their investigation:

1. What are the temporal changes in the isotopic composition of soil water and xylem sap throughout the growing season, and is this behavior unique for each species? 2. Is there evidence for hydrological separation? If so, does that differ between species?

While I think this is an important and well thought and carried out study, I have some concerns:

[Figure]

The discussion is very long and hard to read. While I appreciate the detail, especially by using a review- like approach to discuss the results, I feel the main message is buried under too much detail. I would suggest the authors try and cut the discussion to half the length and keep their focus on the data they worked with, or try and combine Results and Discussion for the first two points (i.e. 4.1 and 4.2) and add the review part (i.e. 4.3) as a discussion/Conclusion section. The reader would benefit a great deal and it would separate the review section clearer from the discussion. For example, the first discussion point addresses the temporal changes in isotopic composition in both soil and xylem. While the authors are doing a good job in describing data from relevant publications, they repeat some the results (e.g. L291ff, L308ff, 336ff) and fail to provide a solid interpretation, which makes this section seem unstructured and not to the point. I understand the question to be answered with this section was a "what"-question, thus indicating a descriptive answer, but the whole manuscript would benefit in my opinion, from a "why"-question, which the authors then later try to provide with the third part of the discussion (i.e. 4.3) in a review like form. I encourage the authors to try and restructure the discussion to one (or another) of the above mentioned forms.

Also, I would encourage the authors to move away from trying to prove the ecohydro-logical separation idea wrong and move towards a solid interpretation of their data (i.e. what causes the offsets between xylem and soil water, and therewith also include a plant focused perspective (i.e. fractionation during water uptake? Fractionation during water transport? Interaction with stored water domains?) much like they tried in the last point of the discussion. That would enable them to formulate clear and concise questions and recommendations for future investigations.

Specific comments:

ABSTRACT:

Include one or two sentences about the most likely explanation at the end of the abstract (if the word count does not allow this, maybe cut one of the introductory sen-

tences).

INTRODUCTION:

L37 include Brooks et al. 2010 with the mentioning of the "two water worlds" hypothesis. Their publication introduces the idea before McDonells et al. 2014 publication.

STUDY AREA AND METHODS

2.3

L104 were the same trees cored five times? How did you manage to extract five cores from the same height? Please elaborate.

2.4

L115 How often were these samples taken? Please add. Also, how long were the samples stored in Ziploc bags before measurements? Please discuss the concerns raised by Herbstritt et al. (2014) and Hendry et al. (2015) regarding potential water losses using ziplog bags in this context.

L118 how far away were the lysimeters to the trees cored for xylem sap? Why did you not use trees close to the lysimeters? Also, make sure you use the word tensiometer or lysimeter or suction cup consistently when talking about the mobile water fraction throughout the manuscript.

L126 the url does not work. Also, this section reads incomprehensible, please try to clarify.

2.5

For all three water pools (bulk soil, mobile, xylem) different methods for water extractions and measurements were used. Why is that? The data seem complicated enough and a common method would at least provide the same methodological artifact for all three pools. Please elaborate.

L159 Please read and discuss the Benettin et al. (2018) publication in this context. They provide solid concerns about best fit regression analysis of samples with regard to the LMWL. The implications could change the interpretations of your results, please also check in folloing sections of the manuscript.

RESULTS

L204 ff and Fig.3b) please indicate if the samples plotting to the right of the LMWL are bulk soil water samples from the summer (expected high evaporation fractionation) or not.

L210 please discuss this in relation to the different extraction/measuremtn techniques

L213 if the bulk soil samples was collected in 5cm increments as indicated in MM 2.4 why not compare the soil data from 5-15 cm instead of 0-15? Isooptic enrichment is expected to be highest in the upmost soil layers, creating a negative lc-excess.

Fig. 4 I find the combination of different colours and symbols is confusing. If I understand the figure right, neither would be necessary since facets were used to indicate different sampling timepoints. I suggest using one colour and symbol and then differentiating with solid and unfilled symbols. Also, please make sure that the axes have the same range and tickmarks. And I think one could benefit from a vertical line indicating a 0 lc-excess.

Generally, when printing the figures, the y axis title is not printed. I don't know if that's due to the figure resolution or format, but it might be worth checking.

DISCUSSION: I don't have specific comments for the discussion at this point. Please consider my suggestions above or if you can find a better solution, that's also great. I would be happy to read the manuscript again.

References used in this review:

Benettin P, Volkmann THM, Freyberg J von, Frentress J, Penna D, Dawson TE, and

Kirchner JW 2018. Effects of climatic seasonality on the isotopic composition of evaporating soil waters. Hydrol. Earth Syst. Sci. 22: 2881–2890.

Hendry MJ, Schmeling E, Wassenaar LI, Barbour SL, and Pratt D 2015. Determining the stable isotope composition of pore water from saturated and unsaturated zone core: Improvements to the direct vapour equilibration laser spectrometry method. Hydrol. Earth Syst. Sci. 19: 4427–4440.

Herbstritt B, Limprecht M, Gralher B, and Weiler M 2014. Effects of soil properties on the apparent water-vapor isotope equilibrium fractionation: Implications for the headspace equilibrium method., p. Albert-Ludwigs-Univ. Freiburg i. Breisgau.

---

## Referee Comment (RC2) · Anonymous Referee #2 · 25 Jan 2021

The authors studied the isotopic dynamics in gross precipitation, bulk soil water, mobile soil water and xylem water of four tree species, observed in a northern mixed forest (Ontario, Canada) during the growing season 2016. They put their results in context with the two water worlds / ecohydrological separation hypothesis.

The manuscript presents a well carried out study and shows a nice data set. The study fits well within the scope of HESS, however, it needs some revision. In particular, the discussion is too long and could benefit from trimming and condensing. The discussion is generally a good review of current literature, however, it is too detailed and distracts too much from the authors' main findings and thereby fails to highlight and explain

the observed differences in bulk soil water, mobile soil water and xylem water. The discussion should focus more on the authors' main results and their interpretation, and for this interpretation, it would help a lot to integrate the points in 4.3 earlier.

It would be also interesting to dig a bit deeper into why xylem water differed between species, pointing more at possible different species' strategies in water use (not just rooting depth) and their influence on xylem water, such as water storage in trunks /other plant compartments, different water use (water spender vs. water saver), dormancy, etc.

In addition, it should be discussed more in detail how the different methods applied could have affected the results. Lysimeter vs. equilibration technique vs. cryogenic extraction. IRMS vs. ICOS (2 different analyzers).

Also, there are many figures (8 figures). Maybe some figures could be moved to the appendix? It would be good to make clearer that there is some overlap in data with a previous study of Snelgrove (2019). (Table 1: tree information, Fig. 6: mobile soil water, SWC data of sites He and Or/Pw?).

I added my line-by-line notes as attachment.

I am looking forward to reading the manuscript again!

Please also note the supplement to this comment:
https://hess.copernicus.org/preprints/hess-2020-592/hess-2020-592-RC2-supplement.pdf

**Supplement:**

ABSTRACT

Line 17-18: You write that xylem water and bulk soil water deviate from LMWL, but you do not explain explicitly in what direction. Instead you put them in relative context. Maybe mention the phrase in line 18 "with xylem water …" later?

Line 22-23: The soil depth may constrain differences in rooting depths but not necessarily in root water uptake depths.

STUDY AREA AND METHODS

Line 113: how many samples in specific (range, average)?

Line 118: tension lysimeters: which brand?

Line 119: why "trees that were not used for xylem water sampling"? And how far are these trees from the other trees?

Line 126: reference

Line 148: more details on the method applied here, please. How were those analyzers calibrated (IRMS, 2 ICOS)? Did you make a cross-comparison of these analyzers?

Line 165: Did you consider dependencies of samples since you always sample the same trees? Did you check on criteria of ANOVA (normal distribution of residuals, homogeneity of variances)?

RESULTS

Line 176: State time range of growing season.

Line 180. Where did you define / explain total water depth / soil water depth?

Figure 2: do you also have data from before June 2016?

Fig 3a and b: you could add precipitation/ soil water, e.g. $\delta 2H$ precip and $\delta 18O$ precip resp. $\delta 2H$ water and $\delta 18O$ water, equation for LMWL

Figure 3c: Please, explain why you summarize 0-15 cm and 30+, maybe offset less big if only 35-45 cm? And 5-15? Which soil areas do lysimeters see?

Line 210: maybe reference to figure already here: "given tree species (Fig 3c)"

Figure 2: How did you determine soil water depth of upper 0.5 m?

Figure 4 uses colors for different periods that have been used before for tree species (use rather none? Or colours of species? Or completely different colours?)

Generally, it might be better not to use the same colours for soil values as for plant values.

The x-axes differ which is not ideal for comparison. Since the y-axis of each plot is the same, you could consider removing the space between plots.

Figure 5: You could again add xylem, $\delta 2H$ xylem. Also here plots share y-axes.

Figure 8: This figure has many colours. But I do understand that you are limited in colours here. You could use different patterns? Or also use grey/white instead of switching to a completely different colour (Or,Pw).

Fig. 8: big differences again within species. How come? Can you explain these differences maybe by tree traits (Table 1)?

DISCUSSION

Line 300: intra-specific as well

Line 350: Give your values.

Line 353: SWCs were relatively large?

I noticed that in the text it says Snelgrove (2019), in the references it is 2020.

Figure 5: big scatter within species as well.

In the figure legends LWML is local mean water line, … meteoric …

Figure 6: typo soili

Figure 6: Just to clarify: do you show here the bulk soil water average, or per depth?

---

## Author Comment (AC1) · 11 Feb 2021

Response to reviewer comments

Reviewer 1 The study by Snelgrove et al. investigates if ecohydrologic separation was possible in a northern mixed forest in Ontario, Canada. Their study design is built to assess the co-evolution of mobile-, bulk soil- and xylem water isotopic compositions during the year 2016. They formulate two questions to be considered during their investigation: 1. What are the temporal changes in the isotopic composition of soil water and xylem sap throughout the growing season, and is this behavior unique for each species? 2. Is there evidence for hydrological separation? If so, does that differ

between species? While I think this is an important and well thought and carried out study, I have some concerns:

The discussion is very long and hard to read. While I appreciate the detail, especially by using a review- like approach to discuss the results, I feel the main message is buried under too much detail. I would suggest the authors try and cut the discussion to half the length and keep their focus on the data they worked with, or try and combine Results and Discussion for the first two points (i.e. 4.1 and 4.2) and add the review part (i.e. 4.3) as a discussion/Conclusion section. The reader would benefit a great deal and it would separate the review section clearer from the discussion. For example, the first discussion point addresses the temporal changes in isotopic composition in both soil and xylem. While the authors are doing a good job in describing data from relevant publications, they repeat some the results (e.g. L291ff, L308ff, 336ff) and fail to provide a solid interpretation, which makes this section seem unstructured and not to the point. I understand the question to be answered with this section was a "what"-question, thus indicating a descriptive answer, but the whole manuscript would benefit in my opinion, from a "why"-question, which the authors then later try to provide with the third part of the discussion (i.e. 4.3) in a review like form. I encourage the authors to try and restructure the discussion to one (or another) of the above mentioned forms.

Also, I would encourage the authors to move away from trying to prove the ecohydrological separation idea wrong and move towards a solid interpretation of their data (i.e. what causes the offsets between xylem and soil water, and therewith also include a plant focused perspective (i.e. fractionation during water uptake? Fractionation during water transport? Interaction with stored water domains?) much like they tried in the last point of the discussion. That would enable them to formulate clear and concise questions and recommendations for future investigations.

Response: We appreciate the Reviewer's suggestions. We propose to modify the Discussion to reduce its length and to focus on the key findings of our study. As the Reviewer suggests, this can be addressed in part by removing the repetition of results

that they noted. We will also remove the material related to the use of best-fit regression lines for the xylem water samples in dual isotope space to estimate the isotopic composition of the source precipitation from the Methods, Results and Discussion. As the reviewer notes (and as is demonstrated in the Benettin et al. HESS 2018 paper that they drew our attention to), there are serious concerns regarding this approach to estimating the source precipitation isotopic composition. We propose to include a short section in the Discussion (based on the Benettin et al. evaporation lines shown in their calculations) indicating that evaporation of xylem water through the tree bark cannot be excluded as a potential cause of our xylem water isotopic composition results.

Specific comments: ABSTRACT: Include one or two sentences about the most likely explanation at the end of the abstract (if the word count does not allow this, maybe cut one of the introductory sentences).

Response: This change will be made.

INTRODUCTION: L37 include Brooks et al. 2010 with the mentioning of the "two water worlds" hypothesis. Their publication introduces the idea before McDonells et al. 2014 publication.

Response: This change will be made.

STUDY AREA AND METHODS 2.3 L104 were the same trees cored five times? How did you manage to extract five cores from the same height? Please elaborate.

Response: Yes, the same trees were cored five times within a small vertical range (a few cm) at breast height diameter. We will provide information on how this was done in the revised version of the paper.

2.4 L115 How often were these samples taken? Please add. Also, how long were the samples stored in Ziploc bags before measurements? Please discuss the concerns raised by Herbstritt et al. (2014) and Hendry et al. (2015) regarding potential water losses using ziplog bags in this context.

Response: This information was provided in the original submission (soil samples were taken concurrent with the xylem water samples – line 113). Samples were stored for no more than 2 weeks prior to analysis. The Hendry et al. (2015) paper suggests that water losses should therefore not be an issue. We will note this in the revised version of the paper.

L118 how far away were the lysimeters to the trees cored for xylem sap? Why did you not use trees close to the lysimeters? Also, make sure you use the word tensiometer or lysimeter or suction cup consistently when talking about the mobile water fraction throughout the manuscript.

Response: This was an error in the original submission – in fact, the lysimeters were placed adjacent to trees sampled from xylem water. This will be corrected in the revised version of the paper. We meant to use "tension lysimeter" consistently throughout the manuscript, and will check to ensure that this is the case.

L126 the url does not work. Also, this section reads incomprehensible, please try to clarify.

Response: The url will be corrected (a space had been inadvertently inserted). The sentence will be reworded in the revised version of the paper.

2.5 For all three water pools (bulk soil, mobile, xylem) different methods for water extractions and measurements were used. Why is that? The data seem complicated enough and a common method would at least provide the same methodological artifact for all three pools. Please elaborate.

Response: We will include references to our previous work that shows that the direct equilibrium method for extracting bulk soil water gives similar results to cryogenic extraction for the type of soils that we examined (e.g. low clay content, low organic matter content). Further, the same instrument was used to measure the isotopic composition of the mobile and bulk soil water samples. We will also note in the revised version of

the paper that the ICOS instrument was cross correlated with the IRMS.

L159 Please read and discuss the Benettin et al. (2018) publication in this context. They provide solid concerns about best fit regression analysis of samples with regard to the LMWL. The implications could change the interpretations of your results, please also check in folloing sections of the manuscript.

Response: We appreciate that the Reviewer drew our attention to this paper. As noted above, we will remove the material related to the use of best-fit regression lines for the xylem water samples in dual isotope space to estimate the isotopic composition of the source precipitation from the Methods, Results and Discussion. Instead, we will include a short section in the Discussion (based on the Benettin et al. evaporation lines shown in their calculations) indicating that evaporation of xylem water through the tree bark cannot be excluded as a potential cause of our xylem water isotopic composition results.

RESULTS L204 ff and Fig.3b) please indicate if the samples plotting to the right of the LMWL are bulk soil water samples from the summer (expected high evaporation fractionation) or not.

Response: We propose to include a new panel in Figure 3 to indicate that most bulk soil water samples plotting to the right of the LMWL were taken in mid-summer.

L210 please discuss this in relation to the different extraction/measuremtn techniques

Response: The same instrument was used to measure the isotopic composition of the mobile and bulk soil water samples. As note earlier, the ICOS instrument was cross correlated with the IRMS.

L213 if the bulk soil samples was collected in 5cm increments as indicated in MM 2.4 why not compare the soil data from 5-15 cm instead of 0-15? Isooptic enrichment is expected to be highest in the upmost soil layers, creating a negative lc-excess.

Response: We thank the Reviewer to pointing this out. We will revise Figure 3c to

compare the mobile soil water samples taken at 10 and 30 cm depths with bulk soil water samples from 5-10 cm and 10-15 cm and 25-30 cm and 30-35 cm depths (we have done this, and the revision does not alter our conclusions).

Fig. 4 I find the combination of different colours and symbols is confusing. If I understand the figure right, neither would be necessary since facets were used to indicate different sampling timepoints. I suggest using one colour and symbol and then differentiating with solid and unfilled symbols. Also, please make sure that the axes have the same range and tickmarks. And I think one could benefit from a vertical line indicating a 0 lc-excess.

Response: We will revise Figure 4 along the lines suggested by the Reviewer.

Generally, when printing the figures, the y axis title is not printed. I don't know if that's due to the figure resolution or format, but it might be worth checking.

Response: We are not sure why this might have occurred. This may have been an issue arising from how the pdf of our manuscript was generated and exactly what software is used for viewing. We note that Reviewer 2 did not comment on this issue.

DISCUSSION: I don't have specific comments for the discussion at this point. Please consider my suggestions above or if you can find a better solution, that's also great. I would be happy to read the manuscript again.

Response: As we noted earlier, we will revise the Discussion along the lines suggested by the Reviewer in order to reduce its length and improve its focus.

References used in this review: Benettin P, Volkmann THM, Freyberg J von, Frentress J, Penna D, Dawson TE, and Kirchner JW 2018. Effects of climatic seasonality on the isotopic composition of evaporating soil waters. Hydrol. Earth Syst. Sci. 22: 2881–2890. Hendry MJ, Schmeling E, Wassenaar LI, Barbour SL, and Pratt D 2015. Determining the stable isotope composition of pore water from saturated and unsaturated zone core: Improvements to the direct vapour equilibration laser spectrometry

method. Hydrol. Earth Syst. Sci. 19: 4427–4440.

Herbstritt B, Limprecht M, Gralher B, and Weiler M 2014. Effects of soil properties on the apparent water-vapor isotope equilibrium fractionation: Implications for the headspace equilibrium method., p. Albert-Ludwigs-Univ. Freiburg i. Breisgau.

---

## Author Comment (AC2) · 11 Feb 2021

Response to Reviewers

Reviewer 2 The authors studied the isotopic dynamics in gross precipitation, bulk soil water, mobile soil water and xylem water of four tree species, observed in a northern mixed forest (Ontario, Canada) during the growing season 2016. They put their results in context with the two water worlds / ecohydrological separation hypothesis. The manuscript presents a well carried out study and shows a nice data set. The study fits well within the scope of HESS, however, it needs some revision. In particular, the discussion is too long and could benefit from trimming and condensing. The discussion

is generally a good review of current literature, however, it is too detailed and distracts too much from the authors' main findings and thereby fails to highlight and explain the observed differences in bulk soil water, mobile soil water and xylem water. The discussion should focus more on the authors' main results and their interpretation, and for this interpretation, it would help a lot to integrate the points in 4.3 earlier.

Response: We thank the Reviewer for these suggestions. As noted in our response to the comments of Reviewer 1 regarding the Discussion, we propose to reduce its length and improve its focus.

It would be also interesting to dig a bit deeper into why xylem water differed between species, pointing more at possible different species' strategies in water use (not just rooting depth) and their influence on xylem water, such as water storage in trunks /other plant compartments, different water use (water spender vs. water saver), dormancy, etc.

Response: We thank the Reviewer for these suggestions. The revised version of the paper will acknowledge that inter-specific differences in water use strategies may assist in explaining the xylem water isotopic composition results. The revised version of the paper will also note that detailed information is lacking regarding the water use strategies of the tree species that we studied, and how this should be a focus of further work.

In addition, it should be discussed more in detail how the different methods applied could have affected the results. Lysimeter vs. equilibration technique vs. cryogenic extraction. IRMS vs. ICOS (2 different analyzers).

Response: As we noted in response to a similar point raised by Reviewer 1, we will include references to our previous work that shows that the direct equilibrium method for extracting bulk soil water gives similar results to cryogenic extraction. We will also note in the revised version of the paper that the ICOS instrument was cross correlated with the IRMS.

Also, there are many figures (8 figures). Maybe some figures could be moved to the appendix? It would be good to make clearer that there is some overlap in data with a previous study of Snelgrove (2019). (Table 1: tree information, Fig. 6: mobile soil water, SWC data of sites He and Or/Pw?).

Response: We thank the Reviewer for this suggestion and will move the current Figures 2 and 6 to the appendix of the revised version of the paper. The revised version of the paper will also note that some of the data appeared in Snelgrove et al. (2019); however, we will retain those data since they assist in interpreting the bulk soil water and xylem water isotopic composition results presented in the current paper.

I added my line-by-line notes as attachment. I am looking forward to reading the manuscript again!

Specific comments ABSTRACT Line 17-18: You write that xylem water and bulk soil water deviate from LMWL, but you do not explain explicitly in what direction. Instead you put them in relative context. Maybe mention the phrase in line 18 "with xylem water . . ." later?

Response: This change will be made in the revised version of the paper.

Line 22-23: The soil depth may constrain differences in rooting depths but not necessarily in root water uptake depths.

Response: This point will be made in the revised version of the paper.

STUDY AREA AND METHODS Line 113: how many samples in specific (range, average)?

Response: This information will be provided in the revised version of the paper.

Line 118: tension lysimeters: which brand?

Response: The tension lysimeters were manufactured using Soil Test 2 bar ceramic cups and PVC tubing. This information will be provided in the revised version of the

paper.

Line 119: why "trees that were not used for xylem water sampling"? And how far are these trees from the other trees?

As we noted in response to a similar comment from Reviewer 1, this was an error in the original submission – in fact, the lysimeters were placed adjacent to trees sampled from xylem water. This will be corrected in the revised version of the paper.

Line 126: reference

Response: We assume that the Reviewer is referring to the error with the url address. This will be corrected in the revised version of the paper.

Line 148: more details on the method applied here, please. How were those analyzers calibrated (IRMS, 2 ICOS)? Did you make a cross-comparison of these analyzers?

Response: Yes, we cross-compared the analyzers. This will be noted in the revised version of the paper. Information regarding calibration of the analyzers (including the standards used) will also be provided.

Line 165: Did you consider dependencies of samples since you always sample the same trees? Did you check on criteria of ANOVA (normal distribution of residuals, homogeneity of variances)?

Response: The issue of dependencies will be acknowledged in the revised version of the paper. Nevertheless, we feel that sampling the same tree at different times is a strength of the study (as will be noted in the revised version of the paper), since inter-tree differences in xylem water isotopic composition on a given sampling date would affect the temporal trajectory of the xylem water results that would be obtained by sampling different trees at different times. Our sampling strategy allows us to examine that trajectory for each of the sampled trees. Homogeneity of variances was examined using Levene's test. This will be noted in the revised version of the paper.

RESULTS Line 176: State time range of growing season.

Response: This information will be provided in the revised version of the paper.

Line 180. Where did you define / explain total water depth / soil water depth?

Response: This information was given in Snelgrove et al. (2019), and the reference will be cited in the revised version of the paper.

Figure 2: do you also have data from before June 2016?

Response: No, we do not.

Fig 3a and b: you could add precipitation/ soil water, e.g. $\delta$2H precip and $\delta$ 18O precip resp. $\delta$2H water and $\delta$18O water, equation for LMWL

Response: We feel that the suggested change would make the Figures too cluttered. However, we will provide the equation for the LMWL in the revised version of the paper.

Figure 3c: Please, explain why you summarize 0-15 cm and 30+, maybe offset less big if only 35-45 cm? And 5-15? Which soil areas do lysimeters see?

Response: We thank the Reviewer for making this point. As noted in our response to a similar comment from Reviewer 1, we will revise Figure 3c to compare the mobile soil water samples taken at 10 and 30 cm depths with bulk soil water samples from 5-10 cm and 10-15 cm and 25-30 cm and 30-35 cm depths (we have done this, and the revision does not alter our conclusions).

Line 210: maybe reference to figure already here: "given tree species (Fig 3c)"

Response: The suggested change will be made in the revised version of the paper.

Figure 2: How did you determine soil water depth of upper 0.5 m?

Response: This information was given in Snelgrove et al. (2019), and the reference will be cited in the revised version of the paper.

Figure 4 uses colors for different periods that have been used before for tree species (use rather none? Or colours of species? Or completely different colours?)

Response: We will revise this Figure based on these suggestions as well as those of Reviewer 1.

Generally, it might be better not to use the same colours for soil values as for plant values.

Response: We agree and will revised this Figure accordingly.

The x-axes differ which is not ideal for comparison. Since the y-axis of each plot is the same, you could consider removing the space between plots.

Response: We agree and will revised this Figure accordingly.

Figure 5: You could again add xylem, $\delta$2H xylem. Also here plots share y-axes.

Response: We are not sure what the Reviewer means in the first part of this comment. However, we agree with the second point and will revised this Figure accordingly.

Figure 8: This figure has many colours. But I do understand that you are limited in colours here. You could use different patterns? Or also use grey/white instead of switching to a completely different colour (Or,Pw).

Response: We thank the Reviewer for this comment and will consider using different colours or patterns in the revised version of this Figure.

Fig. 8: big differences again within species. How come? Can you explain these differences maybe by tree traits (Table 1)?

Response: We thank the Reviewer for making this point. However, there do not appear to be consistent differences in the offsets between trees of a given species between sampling times. This point will be made in the revised version of the paper.

DISCUSSION Line 300: intra-specific as well

Response: We agree, and this point will be added in the revised version of the paper.

Line 350: Give your values.

Response: This information will be provided in the revised version of the paper.

Line 353: SWCs were relatively large?

Response: We do not have SWC data for these periods; however, previous work at the study site indicates large SWCs at these times. This will be noted in the revised version of the paper.

I noticed that in the text it says Snelgrove (2019), in the references it is 2020.

Response: The correct date is 2019. This change will be made in the revised version of the paper.

Figure 5: big scatter within species as well.

Response: We agree, and this point will be noted in the revised version of the paper.

In the figure legends LWML is local mean water line, . . . meteoric . . .

Response: Yes, of course. We apologize for the error, which will be corrected in the revised version of the paper

Figure 6: typo soili

Response: This error will be corrected in the revised version of the paper.

Figure 6: Just to clarify: do you show here the bulk soil water average, or per depth?

Response: The Figure shows all of the individual bulk soil water samples.

---

## Author Response (AR1)

Response to editor's report and reviewer comments (responses in ***bold italics***)

Editor

Dear Authors,
Thank you very much for your thorough replies to the reviews. As mentioned by both reviewers, the manuscript would benefit from shortening the discussion and focusing more on the main findings. The revised manuscript version should also include more plant physiological explanations for your tree-species specific differences in the xylem water isotopic composition. In your revised manuscript, the applied extraction methods and their pros and cons should also be discussed in more detail and how they might have affected your results. As reviewer 1 suggested, I would also encourage you to move away from trying to disprove the ecohydrological separation idea but focus more on your actual findings and their interpretation. I am looking forward to your revised manuscript version.
Best,
Natalie Orlowski

***Dear Dr Orlowski:***
***Many thanks for giving us the opportunity to respond to your comments as well as those of the reviewers. We are convinced that the revised manuscript is much clearer now. As indicated in our responses to the reviewers' comments, we have clarified and shortened the Discussion (by ~1000 words) and have focused it on the main study findings. We have included a section in the Discussion that deals explicitly with aspects of plant physiology and their ability to account for inter-specific differences in xylem water composition noted in our work (lines 383-406 of revision). We have also presented more information regarding our extraction and analytical methods (lines 149-150, 151-157 of revision) and have expanded our Discussion of their influence on our results (lines 313-326 of revision). In brief, our research group has investigated this question directly (Sprenger et al. 2018b, b) and found no systematic offset. On this basis, we contend that it is unlikely that methodological issues have a significant influence on our observations. We have also removed all material related to the testing of the ecohydrological separation hypothesis.***
***Best regards,***
***James Buttle, on behalf of all coauthors***

Reviewer 1
The study by Snelgrove et al. investigates if ecohydrologic separation was possible in a northern mixed forest in Ontario, Canada. Their study design is built to assess the co-evolution of mobile-, bulk soil- and xylem water isotopic compositions during the year 2016. They formulate two questions to be considered during their investigation:
1. What are the temporal changes in the isotopic composition of soil water and xylem sap throughout the growing season, and is this behavior unique for each species? 2. Is there evidence for hydrological separation? If so, does that differ between species? While I think this is an important and well thought and carried out study, I have some concerns:

The discussion is very long and hard to read. While I appreciate the detail, especially by using a review- like approach to discuss the results, I feel the main message is buried under too much

detail. I would suggest the authors try and cut the discussion to half the length and keep their focus on the data they worked with, or try and combine Results and Discussion for the first two points (i.e. 4.1 and 4.2) and add the review part (i.e. 4.3) as a discussion/Conclusion section. The reader would benefit a great deal and it would separate the review section clearer from the discussion. For example, the first discussion point addresses the temporal changes in isotopic composition in both soil and xylem. While the authors are doing a good job in describing data from relevant publications, they repeat some the results (e.g. L291ff, L308ff, 336ff) and fail to provide a solid interpretation, which makes this section seem unstructured and not to the point.I understand the question to be answered with this section was a "what"-question, thus indicating a descriptive answer, but the whole manuscript would benefit in my opinion, from a "why"-question, which the authors then later try to provide with the third part of the discussion (i.e. 4.3) in a review like form. I encourage the authors to try and restructure the discussion to one (or another) of the above mentioned forms.

Also, I would encourage the authors to move away from trying to prove the ecohydrological separation idea wrong and move towards a solid interpretation of their data (i.e. what causes the offsets between xylem and soil water, and therewith also include a plant focused perspective (i.e. fractionation during water uptake? Fractionation during water transport? Interaction with stored water domains?) much like they tried in the last point of the discussion. That would enable them to formulate clear and concise
questions and recommendations for future investigations.

*Response:*
*We appreciate the Reviewer's suggestions and have modified the Discussion in order to reduce its length (by ~1000 words) and to focus on the key findings of our study. The revision was partly based on the suggestion of Reviewer 2 to integrate the points made in section 4.3 in the original submission into a new section 4.2 in the revised Discussion. We feel such a section is valuable because of the complexity of the topic and the different explanations that have been put forward in the literature to account for plant – soil water isotopic differences. In addition, the drivers that are examined in this section could all be relevant explanations for our observed plant – soil water isotopic differences, and hence need to be discussed.*

Specific comments:
ABSTRACT:
Include one or two sentences about the most likely explanation at the end of the abstract (if the word count does not allow this, maybe cut one of the introductory sentences).

*Response:*
*This change was made (lines 22 – 27 in revision).*

INTRODUCTION:
L37 include Brooks et al. 2010 with the mentioning of the "two water worlds" hypothesis. Their publication introduces the idea before McDonells et al. 2014 publication.

*Response:*
*This reference was included (line 39 in revision).*

STUDY AREA AND METHODS
2.3
L104 were the same trees cored five times? How did you manage to extract five cores from the same height? Please elaborate.

*Response:*
***Yes, the same trees were cored five times within a small vertical range (a few cm) at breast height diameter. This information has been provided (lines 111-112 in revision).***

2.4
L115 How often were these samples taken? Please add. Also, how long were the samples stored in Ziploc bags before measurements? Please discuss the concerns raised by Herbstritt et al. (2014) and Hendry et al. (2015) regarding potential water losses using ziplog bags in this context.

*Response:*
***Bulk soil samples were taken concurrent with the xylem water samples, and this information has been provided (line 117 in revision). Samples were stored for no more than 2 weeks prior to analysis. The Hendry et al. (2015) paper suggests that water losses should therefore not be an issue. This has now been noted (lines 121-122 in revision).***

L118 how far away were the lysimeters to the trees cored for xylem sap? Why did you not use trees close to the lysimeters? Also, make sure you use the word tensiometer or lysimeter or suction cup consistently when talking about the mobile water fraction throughout the manuscript.

*Response:*
***This was an error in the original submission – in fact, the lysimeters were placed adjacent to trees sampled from xylem water. This has been corrected (lines 126-127). We now use "tension lysimeter" consistently throughout the manuscript and have checked to ensure that this is the case.***

L126 the url does not work. Also, this section reads incomprehensible, please try to clarify.

*Response:*
***The url has been corrected (line 131 in revision). The sentence has been separated into two shorter sentences to improve clarity (lines 132-133 in revision).***

2.5
For all three water pools (bulk soil, mobile, xylem) different methods for water extractions and measurements were used. Why is that? The data seem complicated enough and a common method would at least provide the same methodological artifact for all three pools. Please elaborate.

*Response:*

*We have included references to our previous work that shows that the direct equilibrium method for extracting bulk soil water gives similar results to cryogenic extraction for the type of soils that we examined (e.g. low clay content, low organic matter content) (lines 151-155 of revision). Further, the same instrument was used to measure the isotopic composition of the mobile and bulk soil water samples, and the ICOS instrument was cross correlated with the IRMS. This has now been noted (lines 149-150 of revision).*

L159 Please read and discuss the Benettin et al. (2018) publication in this context. They provide solid concerns about best fit regression analysis of samples with regard to the LMWL. The implications could change the interpretations of your results, please also check in folloing sections of the manuscript.

*Response:*
*We appreciate that the Reviewer drew our attention to this paper. Material related to the use of best-fit regression lines for the xylem water samples in dual isotope space to estimate the isotopic composition of the source precipitation has been removed from the Methods, Results and Discussion. Short sections have been included in the Results (lines 399-243 in revision) and Discussion (lines 384-387) based on the Benettin et al. evaporation lines shown in their calculations to indicate that evaporation of xylem water through the tree bark cannot be excluded as a potential cause of our xylem water isotopic composition results.*

RESULTS
L204 ff and Fig.3b) please indicate if the samples plotting to the right of the LMWL are bulk soil water samples from the summer (expected high evaporation fractionation) or not.

*Response:*
*We have included a new panel in revised Figure 2 to indicate that most bulk soil water samples plotting to the right of the LMWL were taken in mid-summer.*

L210 please discuss this in relation to the different extraction/measuremtn techniques

*Response:*
*The same instrument was used to measure the isotopic composition of the mobile and bulk soil water samples. As noted earlier, the ICOS instrument was cross correlated with the IRMS (lines 149-150 of revision).*

L213 if the bulk soil samples was collected in 5cm increments as indicated in MM 2.4 why not compare the soil data from 5-15 cm instead of 0-15? Isooptic enrichment is expected to be highest in the upmost soil layers, creating a negative lc-excess.

*Response:*
*We thank the Reviewer to pointing this out. Revised Figure 2d compares the mobile soil water samples taken at 10 and 40 cm depths with bulk soil water samples from 5-10 cm and 10-15 cm, and 35-40 cm and 40-45 cm depths, respectively.*

Fig. 4 I find the combination of different colours and symbols is confusing. If I understand the figure right, neither would be necessary since facets were used to indicate different sampling timepoints. I suggest using one colour and symbol and then differentiating with solid and unfilled symbols. Also, please make sure that the axes have the same range and tickmarks. And I think one could benefit from a vertical line indicating a 0 lc-excess.

***Response:***
***We have revised the Figure (new Figure 3) along the lines suggested by the Reviewer.***

Generally, when printing the figures, the y axis title is not printed. I don't know if that's due to the figure resolution or format, but it might be worth checking.

***Response:***
***We believe this may have been an issue arising from how the pdf of our manuscript was generated and exactly what software was used for viewing.***

DISCUSSION: I don't have specific comments for the discussion at this point. Please consider my suggestions above or if you can find a better solution, that's also great. I would be happy to read the manuscript again.

***Response:***
***As we noted earlier, we have revised the Discussion along the lines suggested by the Reviewer in order to reduce its length and improve its focus.***

References used in this review:
Benettin P, Volkmann THM, Freyberg J von, Frentress J, Penna D, Dawson TE, and Kirchner JW 2018. Effects of climatic seasonality on the isotopic composition of evaporating soil waters. Hydrol. Earth Syst. Sci. 22: 2881–2890.
Hendry MJ, Schmeling E, Wassenaar LI, Barbour SL, and Pratt D 2015. Determining the stable isotope composition of pore water from saturated and unsaturated zone core: Improvements to the direct vapour equilibration laser spectrometry method. Hydrol. Earth Syst. Sci. 19: 4427–4440.

Herbstritt B, Limprecht M, Gralher B, and Weiler M 2014. Effects of soil properties on the apparent water-vapor isotope equilibrium fractionation: Implications for the headspace equilibrium method., p. Albert-Ludwigs-Univ. Freiburg i. Breisgau.

Reviewer 2
The authors studied the isotopic dynamics in gross precipitation, bulk soil water, mobile soil water and xylem water of four tree species, observed in a northern mixed forest (Ontario, Canada) during the growing season 2016. They put their results in context with the two water worlds / ecohydrological separation hypothesis.
The manuscript presents a well carried out study and shows a nice data set. The study fits well within the scope of HESS, however, it needs some revision. In particular, the discussion is too long and could benefit from trimming and condensing. The discussion is generally a good review of current literature, however, it is too detailed and distracts too much from the authors' main

findings and thereby fails to highlight and explain the observed differences in bulk soil water, mobile soil water and xylem water. The discussion should focus more on the authors' main results and their interpretation, and for this interpretation, it would help a lot to integrate the points in 4.3 earlier.

*Response:*
*We thank the Reviewer for these suggestions. As noted in our response to the comments of Reviewer 1 regarding the Discussion, we have reduced the length of the Discussion and improved its focus. This has been done in part by integrating much of the material in section 4.3 in the original submission into a new section 4.2 in the revised Discussion.*

It would be also interesting to dig a bit deeper into why xylem water differed between species, pointing more at possible different species' strategies in water use (not just rooting depth) and their influence on xylem water, such as water storage in trunks /other plant compartments, different water use (water spender vs. water saver), dormancy, etc.

*Response:*
*We thank the Reviewer for these suggestions. We acknowledge that inter-specific differences in water use strategies may assist in explaining the xylem water isotopic composition results. We also note that detailed information is lacking regarding the water use strategies of the tree species that we studied, and that this should be a focus of further work (lines 401-406 of revision).*

In addition, it should be discussed more in detail how the different methods applied could have affected the results. Lysimeter vs. equilibration technique vs. cryogenic extraction. IRMS vs. ICOS (2 different analyzers).

*Response:*
*As we noted in response to a similar point raised by Reviewer 1, we have included references to our previous work that shows that the direct equilibrium method for extracting bulk soil water gives similar results to cryogenic extraction for the type of soils that we examined (e.g. low clay content, low organic matter content) (lines 151-155 of revision). Further, the same instrument was used to measure the isotopic composition of the mobile and bulk soil water samples, and the ICOS instrument was cross correlated with the IRMS. This has now been noted (lines 149-150 of revision).*

Also, there are many figures (8 figures). Maybe some figures could be moved to the appendix? It would be good to make clearer that there is some overlap in data with a previous study of Snelgrove (2019). (Table 1: tree information, Fig. 6: mobile soil water, SWC data of sites He and Or/Pw?).

*Response:*
*We thank the Reviewer for this suggestion and have moved Figures 2 and 6 in the original submission to the appendix of the revision as Supplementary Figures 1 and 6. We make reference to Snelgrove et al. (2019) in the revised caption for Table 1 and refer the reader to this paper for greater detail regarding the SWC data (line 194 in revision).*

I added my line-by-line notes as attachment.
I am looking forward to reading the manuscript again!

Specific comments
ABSTRACT
Line 17-18: You write that xylem water and bulk soil water deviate from LMWL, but you do not explain explicitly in what direction. Instead you put them in relative context. Maybe mention the phrase in line 18 "with xylem water …" later?

*Response:*
***This sentence has been reworded to address this point (lines 16-19 of revision).***

Line 22-23: The soil depth may constrain differences in rooting depths but not necessarily in root water uptake depths.

*Response:*
***We agree, and this point has now been made (lines 401-402 of revision).***

STUDY AREA AND METHODS
Line 113: how many samples in specific (range, average)?

*Response:*
***This information has been provided (lines 119-120 of revision).***

Line 118: tension lysimeters: which brand?

*Response:*
***The tension lysimeters were manufactured using Soil Test 2 bar ceramic cups and PVC tubing. This information has been provided (line 127 of revision).***

Line 119: why "trees that were not used for xylem water sampling"? And how far are these trees from the other trees?

***As we noted in response to a similar comment from Reviewer 1, this was an error in the original submission – in fact, the lysimeters were placed adjacent to trees sampled from xylem water. This has been corrected (lines 126-127 of revision).***

Line 126: reference

*Response:*
***We assume that the Reviewer is referring to the error with the url address, which has been corrected (line 131 of revision).***

Line 148: more details on the method applied here, please. How were those analyzers calibrated (IRMS, 2 ICOS)? Did you make a cross-comparison of these analyzers?

*Response:*
*Yes, we cross-compared the analyzers. As noted earlier, the ICOS instrument was cross correlated with the IRMS (lines 149-150 of revision).*

Line 165: Did you consider dependencies of samples since you always sample the same trees? Did you check on criteria of ANOVA (normal distribution of residuals, homogeneity of variances)?

*Response:*
*The issue of dependencies has been acknowledged (lines 173-175 of revision). We have noted (lines 175-177 of revision) that we feel that sampling the same tree at different times is a strength of the study, since inter-tree differences in xylem water isotopic composition on a given sampling date would affect the temporal trajectory of the xylem water results that would be obtained by sampling different trees at different times. Homogeneity of variances was examined using Levene's test (line 171 of revision).*

RESULTS
Line 176: State time range of growing season.

*Response:*
*This information has been provided (lines 187-188 of revision).*

Line 180. Where did you define / explain total water depth / soil water depth?

*Response:*
*This information was given in Snelgrove et al. (2019) and has now been noted (lines 136-137 of revision).*

Figure 2: do you also have data from before June 2016?

*Response:*
*No, we do not.*

Fig 3a and b: you could add precipitation/ soil water, e.g. $\delta^2H$ precip and $\delta^{18}O$ precip resp. $\delta^2H$ water and $\delta^{18}O$ water, equation for LMWL

*Response:*
*We feel that the suggested change would make the Figures too cluttered. However, we now provide the equation for the LMWL in revised Figure 2.*

Figure 3c: Please, explain why you summarize 0-15 cm and 30+, maybe offset less big if only 35-45 cm? And 5-15? Which soil areas do lysimeters see?

*Response:*

*We thank the Reviewer for making this point. Revised Figure 2d compares the mobile soil water samples taken at 10 and 40 cm depths with bulk soil water samples from 5-10 cm and 10-15 cm, and 35-40 cm and 40-45 cm depths, respectively.*

Line 210: maybe reference to figure already here: "given tree species (Fig 3c)"

*Response:*
*Suggested change has been made (lines 212-214 of revision).*

Figure 2: How did you determine soil water depth of upper 0.5 m?

*Response:*
*This information was given in Snelgrove et al. (2019) and has now been noted (lines 136-137 of revision).*

Figure 4 uses colors for different periods that have been used before for tree species (use rather none? Or colours of species? Or completely different colours?)

*Response:*
*This Figure (revised Figure 3) has been changed based on these suggestions as well as those of Reviewer 1.*

Generally, it might be better not to use the same colours for soil values as for plant values.

*Response:*
*Please see previous response.*

The x-axes differ which is not ideal for comparison. Since the y-axis of each plot is the same, you could consider removing the space between plots.

*Response:*
*We appreciate these points, and this Figure (revised Figure 3) has been changed accordingly.*

Figure 5: You could again add xylem, δ2H xylem. Also here plots share y-axes.

*Response:*
*We are not sure what the Reviewer means in the first part of this comment. However, we agree with the second point and have revised the Figure (revised Figure 4) accordingly.*

Figure 8: This figure has many colours. But I do understand that you are limited in colours here. You could use different patterns? Or also use grey/white instead of switching to a completely different colour (Or,Pw).

*Response:*
*We thank the Reviewer for this comment and have used a grey scale in the revised Figure 6.*

Fig. 8: big differences again within species. How come? Can you explain these differences maybe by tree traits (Table 1)?

*Response:*
**We thank the Reviewer for making this point. However, there do not appear to be consistent differences in the offsets between trees of a given species between sampling times. This point has been noted (lines 265-267 of revision).**

DISCUSSION
Line 300: intra-specific as well

*Response:*
**We agree, and this point has been made (line 285 of revision).**

Line 350: Give your values.

*Response:*
**We feel that the offset values we observed can be readily obtained from revised Figure 6. Therefore no change has been made.**

Line 353: SWCs were relatively large?

*Response:*
**We do not have SWC data for these periods; however, previous work at the study site indicates large SWCs at these times. We now provide a reference to support this point (line 343 of revision).**

I noticed that in the text it says Snelgrove (2019), in the references it is 2020.

*Response:*
**The correct date is 2019. This change has been made throughout the revised version of the paper.**

Figure 5: big scatter within species as well.

*Response:*
**We agree; however, intra-specific variation in xylem water isotopic composition on a given sampling date was similar to that observed in previous work (lines 234-238 of revision).**

In the figure legends LWML is local mean water line, … meteoric …

*Response:*
**We apologize for the error, which has been corrected throughout.**

Figure 6: typo soili

*Response:*
*This error has been corrected (revised Supplementary Figure 2).*

Figure 6: Just to clarify: do you show here the bulk soil water average, or per depth?

*Response:*
*The Figure shows all of the individual bulk soil water samples. This has been noted in the caption for revised Supplementary Figure 2.*

---

## Author Response (AR2)

Response to editor's decision

Editor
Comments to the Author:
Dear Authors,
Thank you very much for your revised manuscript version. It has improved a lot and you have replied to the reviewer's comment in great detail.
I would suggest one last change to be made: Could you please replace the bright green color in all the figures (e.g. for the left panels of the figure on p.16)?
Some figures (and their legends) are a bit small, please pay attention to this during the article's post-processing.
Best regards,
Natalie Orlowski

*Dear Dr Orlowski:*
*We have revised the appropriate Figures as requested.*
*Best regards,*
*James Buttle, on behalf of all coauthors*